# Benefits of Using Dapsone in Patients Hospitalized with COVID-19

**DOI:** 10.3390/vaccines10020195

**Published:** 2022-01-26

**Authors:** Badar A. Kanwar, Asif Khattak, Jenny Balentine, Jong Hoon Lee, Richard E. Kast

**Affiliations:** 1Department of Intensive Care Unit and Neonatal Intensive Care, Hunt Regional Hospital, Greenville, TX 75401, USA; akhattak@huntregional.org (A.K.); jbalentine@huntregional.org (J.B.); 2Science and Research Center, Seoul National University College of Medicine, Seoul 03080, Korea; 3IIAIGC Study Center, Burlington, VT 05408, USA; richarderickast@gmail.com

**Keywords:** ARDS, COVID-19, dapsone, NLRP3

## Abstract

Since the start of the SARS-CoV-2 pandemic, refractory and relentless hypoxia as a consequence of exuberant lung inflammation and parenchymal damage remains the main cause of death. We have earlier reported results of the addition of dapsone in this population to the standard of care. We now report a further chart review of discharge outcomes among patients hospitalized for COVID-19. The 2 × 2 table analysis showed a lower risk of death or discharge to LTAC (Long term acute care) (RR = 0.52, 95% CI: 0.32 to 0.84) and a higher chance of discharge home (RR = 2.7, 95% CI: 1.2 to 5.9) among patients receiving dapsone compared to those receiving the usual standard of care. A larger, blinded randomized trial should be carried out urgently to determine if dapsone indeed improves outcomes in COVID-19.

## 1. Introduction

Vaccines have been tremendously successful in preventing COVID-19. However, there are still numerous unvaccinated individuals in the USA and across the globe, and fatal breakthrough cases do occasionally occur. Standard COVID-19 treatment is evolving.

There are only two approved or commonly used medications for patients hospitalized with COVID-19—dexamethasone and remdesivir. Despite their use as a standard of care and intensive supportive care, multiple forms of mechanical ventilation do not always prevent fatal outcomes [1].

The ferocity of the ongoing COVID-19 epidemic has stressed hospitals worldwide to the point of interfering with care for other illnesses. What are we to do while awaiting an infrastructure for executing rapid trials in an epidemic setting [2,3]? For centuries physicians have answered, to paraphrase former Secretary of Defense Donald Rumsfeld, “[we] go to war [with disease] with the [medicines we] have, not the [medicines we] might want or wish to have at a later time”.

Therefore, in response to requests from patients and their families, we acceded to their entreaties “to try anything that might help” by discussing the potential risks and possible benefit of several untested, unproven treatments and anecdotal reports that have been published recently. This includes the published discussions of experience with, and rationale for, dapsone in the COVID-19 disease [4,5,6,7,8,9,10,11,12].

Casual observations reported earlier provided hints of potential survival benefit from adding dapsone to the standard of care for patients hospitalized with COVID-19 [12]. We reported suspicions that oxygenation and survival after 24–48 h with standard treatment might be better with dapsone [12]. Here we report another review of dapsone use at the Hunt Regional Medical Center. This indicated again that dapsone might offer benefits during COVID-19 treatment.

We hypothesized that dapsone, due to its anti-inflammatory and neutrophil migration inhibitory properties, could be effective in COVID-19-induced ARDS [4,5]. In an earlier small study, we found a survival benefit from adding dapsone to usual care for patients hospitalized with COVID-19 [12]. We reported improvements in oxygenation and survival after 24–48 h with standard treatment plus dapsone [12]. Here we report our further experiences showing dapsone offering benefits during COVID-19 treatment.

## 2. Method

The initiation of dapsone administration began in November of 2020, initially as an off-label treatment in an effort to treat seriously ill patients in the absence of approved treatments. After a few patients seemed to have responded, the Hunt Regional Medical center administration and medical staff called for a pause in prescribing dapsone. The COVID-19 committee was assigned with the task to review the safety and soundness of scientific evidence, as well the ethical aspect of dapsone treatment. After its review, the committee recommended that a protocol needed to be in place for an off-label use of dapsone and to monitor its side effects by any offering physician. All patients were required to have informed consent for dapsone use with an explanation that it is an off-label use and is not FDA approved for use in COVID-19. While the review was being conducted and the protocol developed and reviewed, there was a time window when no one was given dapsone at the hospital. Patients admitted during this time constitute most of the patients in the non-dapsone group used for this report. Since the comparison (no dapsone) group consisted of a retrospective review of existing records, informed consent was not required for those patients.

After approval by the COVID-19 committee, only one physician decided to offer dapsone to all consecutive COVID-19 positive patients on supplemental oxygen, and therapy was administered orally for all patients who gave consent. Only two patients declined to give consent for dapsone treatment.

In the overall complete analysis set, the median age was 64 years, 58% were men, and 42% were women; the most common risk factors for worsening hypoxia and death were CRP over 150 (in 45%), and patients requiring >10 L/min oxygen on admission (37%). The baseline characteristics of the two groups were similar (Table 1). While more patients in the dapsone group had an initial CRP > 150 mg/L (normal 1–3 mg/L), the numbers are too small to assess for significance. By typical measures such as initial CRP and oxygen requirement, patients in the dapsone group were not less sick than those in the control group.

Patients received dapsone 100 mg once to twice daily orally, depending on clinical severity. Along with dapsone, patients received cimetidine 400 mg three times daily to diminish dapsone-related methemoglobinemia [13]. Daily methemoglobin levels were also monitored, and if levels exceeded 11% or higher, dapsone was stopped (three of 30 patients). Due to the unavailability of sufficiently rapid G6PD deficiency testing, reticulocyte counts were monitored for at least 72 h as a surrogate for hemolysis. There were no occurrences of hemolytic anemia.

## 3. Results

Sixteen of thirty patients who received dapsone were discharged home requiring less than 8 L oxygen/min—most less than 5 L oxygen/min; only 6 of 30 patients who did not receive dapsone were discharged home (*p* < 0.001) (Table 2).

## 4. Discussion

Hunt Regional Medical Center in Greenville, Texas, is a very small community hospital in a rural setting. It lacks the basic infrastructure to carry out any randomized trial. Therefore, it decided to only let its medical staff use dapsone as an off-label drug.

We wanted to determine whether what we saw at the bedside would really stand up to the test of science. Therefore, we picked up a comparative group retrospectively from approximately the same time, who was just treated with standard of care.

We acknowledge that our report has several shortcomings. It is neither a pre-designed randomized, blinded and placebo control trial, nor a prospective study.

Furthermore, this informal observational data may reflect a real benefit from dapsone and provide a launching pad to design a formal randomized trial and shall help to determine the number of patients to be included in both groups for appropriate effect.

Although immune modulation is a promising therapeutic avenue for improving outcomes for COVID-19, the most effective targets and intervention strategies remain to be found [14]. Clinical outcomes among severely ill patients with COVID-19 are variable with corticosteroids. It is observed during this clinical study that at least two distinct subgroups exist among COVID-19-induced ARDS. There is a group that responds to steroids alone and improves [15]. The other subgroup continues to worsen despite steroids. It is this group in which the addition of dapsone we think may have helped and reduced mortality.

Our report suggests the other plausible molecular mechanisms as follows:

Prior research in neutrophilic dermatoses indicates that dapsone can inhibit neutrophil-related tissue destruction without compromising the adaptive immune response—an appealing therapeutic strategy. Dapsone as an inflammasome inhibitor is thought to be in competition with the Nod-like receptor family pyrin domain-containing 3 (NLRP3) [4]. (Figure 1)

The following four mechanisms were suggested (Figure 1 legend) [11].
Myeloperoxidase is a kind of oxidoreductase that catalyzes the chemical reaction of the following response: H_2_O_2_ + Cl^−^ = H_2_O + OCl^−^. Dapsone binds to myeloperoxidase and regulates the production of hypochlorite, thereby reducing the inflammatory response of cells.Oxidative competition. The methionine (Met) residue at position 35 in the Aβ C-terminal domain is critical for neurotoxicity, aggregation, and free radical formation initiated by the peptide [9]. The bicarbonate/carbon dioxide pair cannot stimulate one-electron oxidation mediated by a radical carbonate anion (CO_3_^●−^), which efficiently oxidizes the thioether sulfur of the Met residue to sulfoxide. Instead, CO_3_^●−^ causes the one-electron oxidation of methionine residue to sulfur radical cation (MetS^●+^) [10]. Dapsone blocks the one-electron oxidation.Nucleophilic properties of dapsone compete with NLRP3. ORF8b activates NLRP3 through direct interaction of the leucine-rich repeat domain of NLRP3. Nucleophilic properties of dapsone compete with NLRP3. Dapsone binds to the AT-rich region of the minor groove of DNANucleophilic properties of dapsone compete with Ubiquitin. DDS can compete with the ubiquitination cascade. Cysteine thiols and hydroxyls on serines, threonines, leucines, and tyrosines could also potentially be ubiquitinated by an identical mechanism

## 5. Conclusions

Based upon these data, we feel a larger randomized controlled trial of dapsone for patients hospitalized with COVID-19 is warranted. Given the urgency of the current epidemic and ongoing deaths from COVID-19 plus the relatively benign side effect profile of dapsone, we encourage others to report their experiences with dapsone in COVID-19 or indeed in ARDS of any other origin.

## Figures and Tables

**Figure 1 vaccines-10-00195-f001:**
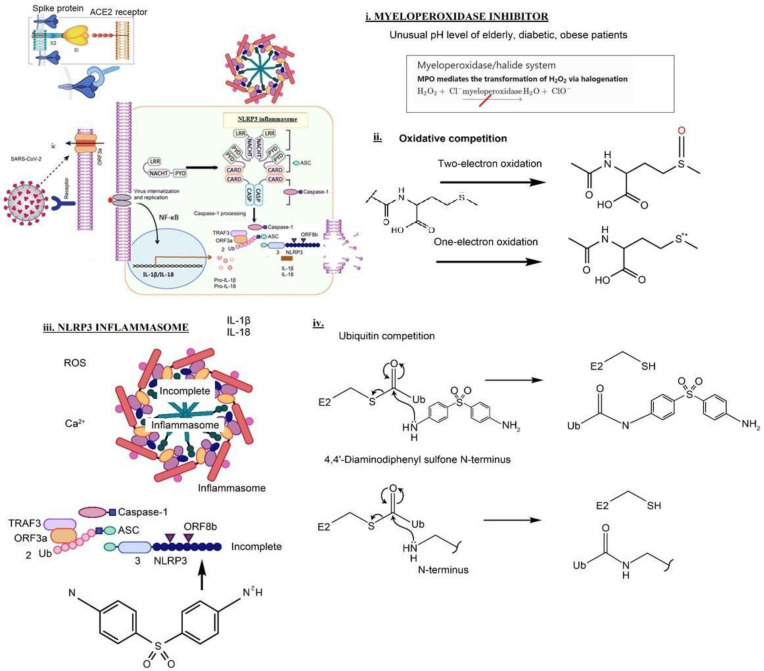
Possible schematic diagram: blocking of SARS-CoV-2 by dapsone [11].

**Table 1 vaccines-10-00195-t001:** Baseline patient characteristics.

	Dapsone + Usual Care (30 Patients)	Usual Care (30 Patients)
Age	62.2 (25–82)	65.8 (26–96)
Sex	17 (57%) male, 13 (43%) female	18(60%) male, 12 (40%) female
Initial CRP > 150	16 (53%) patients	11 (37%) patients
Patients requiring > 10 L/min oxygen on admission	12 (40%) patients	10 (33%) patients
Comorbidities		
Diabetes Mellitus	9	12
Obesity	6	3
COPD	2	5
Malignancy	2	3
Congestive heart failure	4	5
Autoimmune disorders	1	0
Chronic renal disease	4	2

**Table 2 vaccines-10-00195-t002:** Outcomes.

	Dapsone + Usual Care (30 Patients)	Usual Care (30 Patients)
Discharge home	16 patients	6 patients
Died or discharge to LTAC	12 patients	23 patients
Discharge to nursing home or for other medical complication	2 patients	1 patient

LTAC (long term acute care are facilities that specialize in the treatment of patients with serious medical conditions that require care on an ongoing basis but no longer require intensive care or extensive diagnostic procedures).

## Data Availability

The data presented in this study are available in the present article. ClinicalTrials.gov Identifier: NCT04918914.

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
