# Peer review of "Benefits of Using Dapsone in Patients Hospitalized with COVID-19"

_vaccines, 2022, doi:10.3390/vaccines10020195_

Round 1

Reviewer 1 Report

Given the overall impact of the Covid-19 pandemic, novel effective treatments for this disease could be of great importance. Since the onset of the pandemic, multiple "miracle cures" based on repurposing of existing drugs have been proposed based on small poorly controlled trials. These have been met with failure following rigorous testing. While the results reported in this submission are intriguing, the report as submitted, does not satisfy a high enough standard for rigor.

Specific comments

  1. It appears that the Dapsone group were patients treated by one of the authors while the usual care group were those treated by all other physicians at the institution. This immediately introduces an uncontrolled variable. A proper design would be to randomize all patients at the institution who meet entry criteria to either Dapsone or usual care groups.
  2. How were these 30 patients in each group chosen from the total population? Was this study done prospectively or retrospectively? Did the one physician treat half the covid patients? If not, how were the 30 usual care patients selected?
  3. Table 1 lists some characteristics of the patients in both groups. However, we know that outcomes from Covid-19 are highly dependent on various comorbidities such as diabetes and BMI.  A comprehensive comparison of relevant comorbidities between the groups is essential.
  4. The four proposed mechanisms for Dapsone action are interesting, but further substantiation of feasibility based on published literature is needed. The discussion states that the report suggests other molecular mechanisms. Actually, there is nothing in the report that addresses mechanisms in any way.

Reviewer 2 Report

The brief report “Survival benefit of using Dapsone in patients hospitalized 2 with COVID-19”, reported an interesting topic about 30 patients affected by COVID-19 infection treated dapsone, in addition to the standard of care compared to a group 30 patients who did not  receive dapsone.  They found that sixteen of 30 patients who received dapsone were discharged home requiring less than 8L oxygen/min—most less than 5L oxygen/min; only six of 30 patients who did not receive dapsone were discharged home.

The text and contents are understandable..  Minor revision should be considered.

Only few specific concerns:

Please add inclusion and exclusion criteria for selecting patients and some information about clinical characteristic of the patients  treated with dapsone, including radiological findings of ARDS if present, or previous pathologies such as obesity, hypertension, etc, coagulation pattern including D dimer.

Reviewer 3 Report

Title:

Remove the word "Survival: from the title, as it implies that you used survival analysis methods, which you did not.

Abstract:

P1, L13: Replace "Here we report results two groups" with "In this prospective observational study, we report discharge outcomes among two non-randomly selected groups of patients hospitalized for COVID-19."

P1, L14-15: Replace "The comparative group analysis shows survival benefit in dapsone group" with "The 2x2 table analysis showed lower risk of death or discharge to LTAC [spell it out] (RR=0.52, 95%CI: 0.32 to 0.84), and higher chance of discharge home (RR=2.7, 95%CI: 1.2 to 5.9) among patients receiving dapsone compared to those receiving the usual standard of care."

Introduction:

P1, L29: replace "showing survival benefit" with "offering".

Method:

P1, L36: Did the clinician BK offer dapsone to all of his patients, or only those meeting some criteria? What proportion were offered, and what proportion accepted?

P1, L41: Replace "hypxia" with "hypoxia".

P1, L43: I would say the group receiving dapsone were more ill than the control group (i.e., higher proportions had CRP>150, or O2 >10L/min).

Results:

P2, Table 2: What is "LTAC"? Spell it out in a footnote to the table.

I advise appending columns stating the relative risk, and 95% confidence limits of RR (dapsone group vs. control group) for each of the specified outcomes:

Discharge home: RR=2.7, 95%CI: 1.2 to 5.7,

Died or LTAC: RR=0.52, 95%CI: 0.32 to 0.84,

Discharge to nursing home: RR=2.0, 95%CI: 0.2 to 20.9.

Discussion:

P2, L60: I would say that if the group receiving dapsone were more ill than the control group, that would have been a negative bias, and the benefit of dapsone may have been underestimated.

Conclusion:

P4, L97: Replace "this" with "these".

Round 2

Reviewer 1 Report

The authors have provided a response to the critique by adding additional information and changing the description of the method. They have still not adequately addressed how the 30 patients in each group were selected. Were there exactly 30 patients during the period during which Dapsone treatment was paused? If not, how were those 30 patients selected?

Of more concern is the description of the assignment of groups. The origninal description said, "All patients who received dapsone gave written informed consent. One clinician (BK) offered dapsone as adjuvant treatment for patients hospitalized with  COVID-19. As a comparison population, patients cared for by others received usual care  without Dapsone. We compared 30 patients treated with dapsone and standard care with 38 contemporary 30 patients receiving standard care without dapsone.", while the revision stated, "All patients who were COVID-19 positive and on supplemental oxygen were offered  dapsone and only those were excluded who declined to consent for dapsone. Otherwise, no other exclusion or inclusion criteria was used."  These do not seem to be internally consistent. Were all patients offered Dapsone or just those of BK? In general, there appear to be significant internal contradictions in the description of the experimental design.

Also, they now state that the Dapsone group was sicker based on CRP and oxygen >10L/min. Are these statistically significant?

Author Response

Please see upload

Reviewer 3 Report

Abstract:

P1, L59: Replace "standard" with "the usual standard".

P1, L60: Replace "other group also" with "other group were volunteers who also".

Method:

P1, L82: Replace the period after "responded" with a comma.

P1, L87: Replace the period after "physicians" with a comma.

P2, L123, L130-131: These statements do not agree with each other. Please clarify: the control subjects were BK's patients who were not offered dapsone during the specified time period, or they were the patients of other clinicians?

Results:

P3, L185: Append this sentence to the end of the paragraph: " Among patients receiving dapsone in addition to the usual care, there was lower risk of death or discharge to long term acute care (12/30 versus 23/30, relative risk 0.52, 95% confidence interval: 0.32 to 0.84), and higher chance of discharge home (16/30 vs. 6/30, RR=2.7, 95% CI: 1.2 to 5.9) compared to those receiving the usual standard of care."

Supplementary statistics:

Replace "Bad outcome" with "Died or discharged to LTAC".

Replace "Good outcome" with "Discharge home".

Replace "Group 1" with "Usual care".

Replace "Group 2" with "Dapsone + usual care".

Author Response

Please see upload

Round 3

Reviewer 1 Report

The authors have clarified several issues, but others remain unresolved.

  1. This is not really a prospective study. It appears that the data for 25 of the control patients already existed prior to approval of the protocol.
  2. Was this protocol approved by a chartered IRB? This needs to be explicitly stated.
  3. Did the control patients sign informed consent for use of their data. The 25 pre-existing may be exempt since the data already existed, but if 5 patients were entered prospectively, they should be consented.
  4. Were the 5 control patients the first 5 patients who met the inclusion criteria? If not, how were the 5 chosen?
  5. Also, for the 30 Dapsone treated patients, were these the first consecutive patients of BK who consented?  How many patients were offered Dapsone, but did not consent? How was the number 30 chosen for patient enrollment?
  6. Line 56-58; This sentence needs to be edited.
  7. Line 73; please provide route of administration.
  8. In the Methods, please provide a section on statistical analysis.
  9. Line 94; This report does not suggest any potential mechanisms; however, the suggested mechanisms are plausible.
  10. In the conclusions, please state that these data are from a preliminary, nonrandomized trial. Also acknowledge that the fact that the patients in the two groups were treated by different physicians with only BK giving Dapsone is a significant weakness.

Author Response

We truly appreciate reviews comments and suggestions.  We have reviewed our manuscript extensively based upon suggestions.

BK

Comments and Suggestions for Authors

The authors have clarified several issues, but others remain unresolved.

  1. This is not really a prospective study. It appears that the data for 25 of the control patients already existed prior to approval of the protocol.

Response:  We agree our study is not truly a prospective study.  We have edited our manuscript to reflect the real circumstances and possible scientific classification where we feel it fits well.

  1. Was this protocol approved by a chartered IRB? This needs to be explicitly stated.

Response:  Hunt Reginal Medical Center is a very small community hospital where basic infra structure to carry out a trail totally lack.  Hence, a trial couldn’t be carried out.   Manuscript is edited to depict ground realities.

  1. Did the control patients sign informed consent for use of their data? The 25 pre-existing may be exempt since the data already existed, but if 5 patients were entered prospectively, they should be consented.

Response:  25 patients are from pause period and 5 were taken from time immediately prior to start of any dapsone therapy.

  1. Were the 5 control patients the first 5 patients who met the inclusion criteria? If not, how were the 5 chosen?

Response:  Since there were only 25 patients available in pause time.  Other 5 were randomly chosen from immediately prior to dapsone therapy time.

  1. Also, for the 30 Dapsone treated patients, were these the first consecutive patients of BK who consented?  How many patients were offered Dapsone, but did not consent? How was the number 30 chosen for patient enrollment?

Response:  Yes, these are consecutive patients.  Only 2 decline to consent of 32.

  1. Line 56-58; This sentence needs to be edited.
  2. Line 73; please provide route of administration.

Response: Done

  1. In the Methods, please provide a section on statistical analysis.

Response: Please see the supplement.

  1. Line 94; This report does not suggest any potential mechanisms; however, the suggested mechanisms are plausible.

Response:  Done

  1. In the conclusions, please state that these data are from a preliminary, nonrandomized trial. Also acknowledge that the fact that the patients in the two groups were treated by different physicians with only BK giving Dapsone is a significant weakness.

Response:  Included in discussion part.

Also submitting RAW data files for reviewers.

Thanks a lot.

Round 4

Reviewer 1 Report

The authors have made some revisions. Some of these improved the paper, while others added material that is inappropriate for a scientific paper (eg, lines 69-79). More importantly, the authors now acknowledge that the study was performed without an IRB approval. While this reviewer is sympathetic to the limited resources at the small community hospital, this is a fatal flaw. It is possible to contract with an outside IRB for review. The instructions for authors for MDPI journals explicitly state that IRB approval is required for non-exempt human studies. As chair of an IRB (at a community hospital), I can attest that this protocol would require full board review.  

Round 5

Reviewer 1 Report

  • Lines 69-79 are overly melodramatic and have not place in a scientific paper. Please delete.
  • please revise line 150 from "The initiation of dapsone administration began in November of 2020, initially as a desperate 150 effort to save human lives." to "..., initially as an off label treatment in an effort to treat seriously ill patients in the absence of approved treatments."
  • Line 169-170, Please describe how these 5 patients were randomly selected.
  • Please add that "Since the control (no Dapsone) group was consisted of retrospective review of existing records, informed consent was not required.
  • The statements "Medicine has and will always remain more of an art than a science. However, medicine does uses lots of science to a good effect." are not true in reputable medical centers and are an insult to those who have worked so hard for decades to advance medicine. Please delete.

Author Response

Response:

  1. Lines 69-79 deleted
  2. Line 150 revised to reflect the suggestions.
  3. Lines 169-170 edited to remove concerns.
  4. Statement of concern deleted
